# A New Approach in Lipase-Octyl-Agarose Biocatalysis of 2-Arylpropionic Acid Derivatives

**DOI:** 10.3390/ijms25105084

**Published:** 2024-05-07

**Authors:** Joanna Siódmiak, Jacek Dulęba, Natalia Kocot, Rafał Mastalerz, Gudmundur G. Haraldsson, Michał Piotr Marszałł, Tomasz Siódmiak

**Affiliations:** 1Department of Laboratory Medicine, Faculty of Pharmacy, Ludwik Rydygier Collegium Medicum, Nicolaus Copernicus University in Toruń, 85-094 Bydgoszcz, Poland; joanna.pollak@cm.umk.pl; 2Department of Medicinal Chemistry, Faculty of Pharmacy, Collegium Medicum in Bydgoszcz, Nicolaus Copernicus University in Toruń, 85-089 Bydgoszcz, Poland; jac.duleba@gmail.com (J.D.); natalia.kocot@doctoral.uj.edu.pl (N.K.); rafi.mastalerz@gmail.com (R.M.); mmars@cm.umk.pl (M.P.M.); 3Department of Pharmaceutical Technology, Faculty of Pharmacy, Medical Biotechnology and Laboratory Medicine, Pomeranian Medical University in Szczecin, 71-251 Szczecin, Poland; 4Doctoral School of Medical and Health Sciences, Jagiellonian University, Łazarza 16, 31-530 Kraków, Poland; 5Department of Pharmaceutical Biochemistry, Faculty of Pharmacy, Jagiellonian University Medical College, Medyczna 9, 30-688 Kraków, Poland; 6Science Institute, University of Iceland, 107 Reykjavik, Iceland; gghar@hi.is

**Keywords:** octyl-Sepharose CL-4B, octyl-agarose, lipase B from *Candida antarctica*, lipase from *Candida rugosa*, immobilization, (*R*,*S*)-flurbiprofen, kinetic resolution, reactor material, climatic chamber, storage stability, enantioselective esterification, polypropylene

## Abstract

The use of lipase immobilized on an octyl-agarose support to obtain the optically pure enantiomers of chiral drugs in reactions carried out in organic solvents is a great challenge for chemical and pharmaceutical sciences. Therefore, it is extremely important to develop optimal procedures to achieve a high enantioselectivity of the biocatalysts in the organic medium. Our paper describes a new approach to biocatalysis performed in an organic solvent with the use of CALB-octyl-agarose support including the application of a polypropylene reactor, an appropriate buffer for immobilization (Tris base—pH 9, 100 mM), a drying step, and then the storage of immobilized lipases in a climatic chamber or a refrigerator. An immobilized lipase B from *Candida antarctica* (CALB) was used in the kinetic resolution of (*R*,*S*)-flurbiprofen by enantioselective esterification with methanol, reaching a high enantiomeric excess (ee_p_ = 89.6 ± 2.0%). As part of the immobilization optimization, the influence of different buffers was investigated. The effect of the reactor material and the reaction medium on the lipase activity was also studied. Moreover, the stability of the immobilized lipases: lipase from *Candida rugosa* (CRL) and CALB during storage in various temperature and humidity conditions (climatic chamber and refrigerator) was tested. The application of the immobilized CALB in a polypropylene reactor allowed for receiving over 9-fold higher conversion values compared to the results achieved when conducting the reaction in a glass reactor, as well as approximately 30-fold higher conversion values in comparison with free lipase. The good stability of the CALB-octyl-agarose support was demonstrated. After 7 days of storage in a climatic chamber or refrigerator (with protection from humidity) approximately 60% higher conversion values were obtained compared to the results observed for the immobilized form that had not been stored. The new approach involving the application of the CALB-octyl-agarose support for reactions performed in organic solvents indicates a significant role of the polymer reactor material being used in achieving high catalytic activity.

## 1. Introduction

The development of new drugs and their building blocks is an increasingly important part of the pharmaceutical industry. Due to the limited effectiveness of active substances applied as racemic mixtures and the risk of the wrong enantiomer being harmful, the significance of the pure enantiomers of active compounds or their precursors is of vital importance [1,2]. Receiving these enantiomers by chemical synthesis is associated with substantial hurdles, such as low process efficiency, high costs, and the need to use reagents, which may harm the environment. In contrast, the application of enzymatic catalysis favors performing reactions under mild conditions. Moreover, these reactions have advantages over chemical ones, mainly due to low reaction costs and environmental friendliness [3].

Among the biocatalytic reactions, those of pharmaceutical significance are of high interest. One of the most applied groups of enzymes is the lipases (EC 3.1.1.3.). Due to their unique properties, such as action at the phase interface and the ability to change conformation under the influence of the hydrophobic reaction medium, they are widely used in reactions leading to the discovery of new drugs [4,5,6]. A substantial part of biocatalysts from this class possesses a specific polypeptide chain (known as the “lid”) covering their active site. The movement of the lid, resulting from the medium influence, determines the shift of the equilibrium to an open or closed lipase conformation. In the open conformation, the hydrophobic pocket exposes the active site to the medium. However, in the case of the closed form, the hydrophobic part of the lid interacts with the hydrophobic area of the active site of the biocatalysts, while the hydrophilic part of the lid is directed towards the medium. The lid shifts upon contact with a hydrophobic surface (oil drop), and the lipase in the open form is adsorbed onto the hydrophobic area of the drop. The equilibrium change to the open form occurs, and the water/oil interface interaction takes place. This phenomenon is called an interfacial activation [7,8,9].

An important feature of lipases is their enantioselectivity, i.e., the ability to catalyze the resolution of a racemic mixture with a preference for one of the enantiomers [10], which is converted at a much higher rate than the other one (*k_E_*_1_ > *k_E_*_2_) (Figure 1) [11]. Therefore, the kinetic resolution of chiral compounds is an important reaction for obtaining optically pure drugs or their building blocks. In numerous scientific papers, the higher therapeutic activity and/or lower side effects of one enantiomer, compared to the other enantiomer or the racemate, have been presented [12,13,14]. To achieve the optically pure drug or building block, the lipase should be characterized by a high enantioselectivity.

The lipase B from *Candida antarctica* (CALB) is one of the most frequently applied enzymes in obtaining the optically pure enantiomers of medicinal substances [15]. It is characterized by the catalytic triad Ser-105, His-224, and Asp-187 [16,17]. The presence of the so-called “lid”, which allows lipase to change its conformation from open to closed, remains a controversial topic [18]. This enzyme has a very small lid that is unable to fully cover the active site and its closed form is not closed to the same extent as in other lipases [7]. What is essential, the CALB shows high activity at temperatures of 10–60 °C and pH 6–9 [16,17]. One of the frequent applications of the CALB is the kinetic resolution of the derivatives of 2-arylpropionic acid [19,20]. Khiari et al. [21] used the CALB as the catalyst to obtain enantiopure (*S*)-(+)-ibuprofen. This enantiomer was described as having lower side effects than a racemic mixture [22]. The CALB has also been utilized in the kinetic resolution of other non-steroidal anti-inflammatory drugs (NSAIDs). Zdun et al. [23] obtained pure (*S*)-naproxen, which demonstrates ca. 28-fold stronger effect than (*R*)-naproxen, using the reaction catalyzed by this lipase. The CALB was likewise used in the kinetic resolution of (*R*,*S*)-flurbiprofen via esterification [18] (Figure 2). It should be mentioned that the two enantiomers of (*R*,*S*)-flurbiprofen exhibit different activities—(*S*)-flurbiprofen shows anti-inflammatory effects, while the (*R*)-enantiomer has antinociceptive and antiproliferative properties [19]. Furthermore, the CALB has been utilized in the kinetic resolution of the compounds serving as the building blocks for drugs. Spelmezan et al. [24] used the CALB in the resolution of 1-benzo[*b*]thiophen-2-yl-ethanol, the key intermediate for the synthesis of Zileuton, a 5-lipoxygenase inhibitor, applied in asthma therapy. Moreover, the CALB catalyzed the achievement of chiral amines as important pharmaceutical building blocks [25].

The lipase from *Candida rugosa* (CRL), similar to the CALB, belongs to one of the most applied enzymes in obtaining enantiopure drugs or their precursors [17]. Compared with the CALB, the CRL demonstrates high activity in a narrower temperature range (30–50 °C) and at a pH value of 7.0 [26]. The presence of a lid in the CRL structure and therefore also the conformation change from closed to open influenced by the hydrophobic reaction medium is well documented [27]. Moreover, the CRL exhibited enantiopreference towards the (*S*)-enantiomer in the transformation of racemic mixtures [28]. In the pharmaceutical industry, the CRL has been mainly utilized in the kinetic resolution of the drugs belonging to the groups of 2-arylpropionic acid derivatives [15] and antihypertensive drugs [29].

The CALB and CRL demonstrate high catalytic activity and stability. However, the catalytic properties of the enzymes are often limited when used in free form. It should be noted that the development of the biocatalytic studies involving lipases is aimed at improving the catalytic parameters of the tested enzymes while minimizing reaction costs and maintaining environmental friendliness. One of the applied techniques to achieve an increased stability, the convenient handling of the enzyme, reusability, the facile separation of the enzymes from reaction mixtures, and the prevention of enzyme contamination in products is immobilization. Immobilized lipase offers the possibility to catalyze the reactions in a broader temperature and pH range than its free form. Wahab et al. [30] classified the immobilization techniques depending on the process reversibility—the physical (mainly hydrophobic and van der Waals forces) and ionic interactions are perceived as reversible. In turn, immobilization by strong covalent bonding is an example of an irreversible method. In the covalent immobilization method, for stable enzyme bonding, more enzyme movements and rearrangements are required [31]. In recent years, an immobilization method using interfacial activation has become increasingly popular. It should be remarked that this strategy enables the one-step immobilization, purification, hyperactivation, and stabilization of the lipase [30,32].

Agarose is a widely applied material for lipase immobilization via, among others, interfacial activation. This support is described as a biocompatible polysaccharide polymer obtained from marine algae or seaweed, composed of the repeating units of D-galactose and 3,6-anhydro-alpha-L-galactopyranose. Importantly, agarose is a biocompatible, renewable, biodegradable, and mechanically stable gel [30]. The significant possibilities of modifying agarose as a support by adding a plethora of chemical groups (e.g., glyoxyl, octyl, alkyl, aldehyde, and glumatic) can increase the lipase catalytic properties [33]. Octyl-Sepharose CL-4B is the commercial material derived from agarose (cross-linked 4% agarose with an octyl group). The beneficial effect of lipase immobilization on octyl-agarose beads has been documented in numerous manuscripts [17]. However, the effect of agarose beads on lipase activity tested in organic solvents is poorly described [34]. Therefore, the authors attempted to investigate the enantioselectivity of the CALB and CRL immobilized on the octyl-agarose support. Additionally, due to the potentially significant influence of the immobilization conditions on lipase catalytic activity, the impact of this process was also examined. It is worth mentioning that in previous studies [17,35], protocols concerning the immobilization of the CALB and CRL onto octyl-agarose have been introduced. The assessment of the lipolytic activity of the immobilized lipases conducted in an aqueous medium allowed the development of new, optimized catalytic systems.

This paper demonstrates the enantioselectivity of the CALB and CRL immobilized onto an octyl-agarose support in the kinetic resolution of (*R*,*S*)-flurbiprofen in an organic solvent. Furthermore, an optimization of the immobilization process by the study of the influence of different buffers was performed. Additionally, the influence of the reactor material on the activity of the lipases was examined. Extensive research was complemented by investigating the stability of the immobilized lipases by storage under various thermal and humidity conditions (climatic chamber and refrigerator). 

## 2. Results and Discussion

### 2.1. CALB—Optimization of Immobilization

#### 2.1.1. Effect of Buffer pH

The CALB was immobilized on the octyl-agarose support via interfacial activation using 100 mM buffers of different pH values (pH 4 (citrate), pH 7 (phosphate), and pH 9 (Tris base)). The immobilization process and reactions were conducted in polypropylene vials. The enantioselectivity in organic solvent (1,2-dichloropropane—DCP) was tested, and the results were presented as the enantiomeric excess of the substrate (ee_s_), enantiomeric excess of the product (ee_p_), and conversion (c). The received data are demonstrated in Table 1. The amount of the lipase immobilized on the support was determined using the Bradford method.

The results showed that the conversion values increased with the increasing pH value of the buffer used for the lipase immobilization. The highest conversion values were observed when using the Tris base buffer (pH 9). After 24 h of the kinetic resolution of (*R*,*S*)-flurbiprofen by enantioselective esterification with methanol, the *R*-flurbiprofen methyl ester was obtained with an enantiomeric excess of ee_p_ = 89.6 ± 2.0%. The conversion values observed when applying the lipase immobilized at pH 9 were more than 2- (after 48 h) or 4 (after 24 h) -fold higher than those received when the CALB immobilized in the citrate (pH 4) or phosphate (pH 7) buffers were used. Based on these results, it is evident that the pH of the buffer utilized for immobilization greatly affects the parameters of the kinetic resolution in the organic solvent.

It is assumed that the Tris base buffer (alkaline pH) favors reaching the optimal structural conformation of the immobilized lipase (above the lipase isoelectric point—pH 6.0) for its catalytic activity (the enantioselective esterification of flurbiprofen) in an organic solvent. The tested buffer may have a stabilizing effect on the active site of the CALB immobilized on octyl-agarose beads. The high activity in an organic solvent (medium) could be related to the effect of the buffer on the ionization state of the catalytic triad and the interaction between the CALB and the support [36,37]. Additionally, the lipase loading could also have a significant impact on its activity. The lowest amount of immobilized lipase was acquired in the pH 9 buffer (lipase loading at pH 9—30.5 ± 0.6 mg/g support), which may result in improvement in the substrate’s availability for the lipase during catalysis, as well as probably affecting the enzyme aggregation limitation [18]. Lipase loading in the pH 4 buffer was 64.8 ± 0.9 mg/g support and in the pH 7—42.1 ± 1.4 mg/g support. The high loading of the support with biocatalysts may cause enzyme crowding on the support surface [38]. It is noticeable that the other tested buffers may not be conducive to reaching the optimal lipase conformation for performing the kinetic resolution of (*R*,*S*)-flurbiprofen in an organic solvent. The optimization of this parameter seems to be one of the crucial steps in developing a new approach to creating an effective biocatalytic system in an organic medium.

A phosphate buffer (pH 7) has often been used to immobilize the CALB on octyl-agarose beads [39]. In our previous paper [35], we performed the immobilization of this lipase in a citrate buffer at pH 4. It should be noted that studies on the activity of the immobilized CALB on octyl-agarose described in the literature were most often conducted under aqueous conditions [40], whereas in the current study, we evaluated the lipase activity in the enantioselective esterification of (*R*,*S*)-flurbiprofen with methanol in an organic solvent. Analyzing the data, we observed a strong relationship between the pH value of the buffer used for the immobilization and the substrates and medium (aqueous or organic) under which the immobilized enzyme was applied. Therefore, when selecting the optimal pH of the buffer, the chemical structure of the substrates, medium, and the type of reaction should also be considered.

Based on the highest values of the conversion and enantiomeric excess of the product, it was decided that the immobilization protocol with the Tris base buffer would be used for further tests.

The analysis was based on pH values and buffer concentrations, but it should be noted that the ionic strength and the nature of the buffers (chemical composition) used in the procedure may also impact the enzyme activity. Braham et al. [41] described that the presence of a Tris buffer in the procedure of enzyme immobilization on a glyoxyl agarose generates a reduction in the enzyme stability. They suggested that the effect could be linked with the blockage of the aldehyde groups in the support caused by the chemical composition of the Tris buffer. Based on docking studies, Schmidt et al. [42] demonstrated that the binding of the polymeric substrate in a groove located at the protein surface is interfered with by the Tris buffer. In the literature [40], it has been pointed out that for studies involving the use of an immobilized enzyme performed in different buffers, the nature of the buffer significantly influences the stability of the lipase immobilized via interfacial activation and, also that phosphate usually generates negative results in biocatalyst stability, including the CALB. Moreover, it is indicated that the enzyme loading (potentially by intermolecular interaction) and the nature of the buffer, apart from the effect on the enzyme stability, also impact enzyme catalytic activity or specificity [43].

#### 2.1.2. Effect of Buffer Concentration

The CALB immobilization with the application of buffers at pH 9 and different concentrations (50 mM, 100 mM, 300 mM, and 500 mM) were investigated. The immobilization process and enantioselective esterification reactions were conducted in polypropylene vials. The enantioselectivity of the CALB in 1,2-dichloropropane (DCP) was tested, and the results were expressed in terms of the enantiomeric excess of the substrate (ee_s_), enantiomeric excess of the product (ee_p_), and conversion (c). The obtained results are presented in Table 2.

The data presented in Table 2 indicate a great influence of the concentrations of the tested Tris base buffer (pH 9) on the enantioselectivity of the immobilized lipase in the kinetic resolution of (*R*,*S*)-flurbiprofen with methanol. In the case of the studied immobilization conditions, the lowest values of conversion were marked for the lipase immobilized in a buffer of 500 mM, whereas the highest values of conversion were reached when using a buffer of 100 and 300 mM. The conversion values increased with the increase in the concentrations of the buffer (in the tested range), while in the buffer of 500 mM, there was a substantial decrease in the catalytic activity. The analysis of the results indicates that the increase in the enzyme’s enantioselectivity in an organic solvent under the conditions of 100 mM buffer may result from the favorable orientation of the enzyme during the immobilization.

The analysis of the data obtained for the immobilized and free lipase indicates that the catalytic activity of the immobilized lipase after 24 h and 48 h of incubation was approximately 30-fold higher as compared to that of the free form (30.2 ± 0.9% versus 1.0 ± 0.1%, and 38.8 ± 0.5% versus 1.2 ± 0.1%) (Table 2 and Table 3). The reactions involving the free and immobilized lipase were performed in a polypropylene reactor, with DCP as the reaction medium. The results indicate that the use of an approach that includes the optimal selection of the pH and the concentration of the buffer allows for a significant increase in the reaction conversion values compared to the free form of the lipase. Therefore, an appropriate selection of the parameters of the buffers used at the stage of the lipase immobilization on the octyl-agarose support constitutes a very useful tool for optimizing the activity of the biocatalyst in kinetic resolution in an organic solvent.

In our previous paper [35], we extensively discussed, based on our own results and literature data, the influence of the buffer used for immobilization on lipase activity. It should be noted that in the previous publication, we conducted our studies in an aqueous medium (the hydrolysis of olive oil), while in this project we assessed the activity in an organic solvent (enantioselectivity). Our results indicate that the immobilization of the CALB on octyl-agarose in buffers with higher concentrations (regardless of the pH) offers better results in terms of activity in reactions carried out in both aqueous and organic media. However, this relationship may change when different substrates are applied. 

Ions in the solution play an essential role because they influence, among others, the viscosity and the tension of the surface of the aqueous medium, as well as the solubility and surface charge of proteins. Some ions in the buffer can affect the electrostatic interaction between the protein and protein and also between the protein and substrate. It should be noted that the ionic strength-dependent molecular forces are intrinsically controlled by the dissociation and hydration of the ions at the interface [44]. Immobilization on octyl agarose concerns the open form of the lipase and it is not treated as a conventional hydrophobic adsorption. At high ionic strength, immobilization on this support is slower, and what is essential, the biocatalyst maintains its open form even at very high ionic strength [8].

The data presented in this paper constitute another argument for the discussion on the influence of the buffer utilized in the immobilization protocol on the activity of the lipase B from *Candida antarctica* when immobilized on the octyl-agarose.

### 2.2. CALB—The Reactor Material, Reaction Medium, and Lipase Stability Tests

#### 2.2.1. Effect of Reactor Material

To determine the influence of the reactor material on the enantioselectivity of the CALB immobilized on the octyl-agarose support, reactors made from glass and polypropylene were used. Both materials are widely applied in chemical, biological, medical, and pharmaceutical research. Two strategies were developed, the first one based on the utilization of a glass reactor during the immobilization and the subsequent esterification in an organic solvent, while the second one involved the use of the vials made of polypropylene for the immobilization, as well as for the kinetic resolution. The enantioselective esterification of (*R*,*S*)-flurbiprofen with methanol was carried out in two solvents: DCM and DCP. The following parameters were calculated: c, ee_p_, and ee_s_. The obtained results are presented in Table 4.

When investigating the achieved results of the catalytic activity of dry lipase supports (air-dried for 48 h, after immobilization) used in the reactors made of glass and polypropylene, a significant difference in the values of the catalytic parameters was noticeable. The esterification reactions conducted in DCM and catalyzed by the immobilized lipase (pH 7 and 100 mM) resulted in over 8-fold higher conversion values when using the polypropylene reactor as compared with the glass one, whereas when using DCP in the kinetic resolution catalyzed by the immobilized lipase (pH 7; 100 mM), the conversion in the polypropylene reactor became more than 3-fold higher than in the glass reactor. In turn, the application of the lipase (immobilized at pH 9; 100 mM) in the esterification taking place in the polypropylene reactor resulted in an over 9-fold higher activity (conversion) as compared with the glass reactor.

It is believed that when dry octyl-agarose supports are utilized, the polypropylene reactor can promote a greater mobility of the immobilized lipase in an organic medium, whereas the use of a glass reactor probably may reduce the exposure of the immobilized lipase to the substrates, which results in lower values of the catalytic parameters. The observed adhesion to the glass reactor walls of the dry octyl-agarose supports may be of significant importance and could be a challenge in obtaining effective biocatalytic systems in an organic medium. Moreover, it is worth noting that polypropylene applied as a lipase support demonstrates special properties that help in the activation of the mechanism of the lid opening and improve the catalytic efficiency of the immobilized enzymes [45].

Comparing the data obtained for the free and immobilized lipase in the reaction carried out in DCM, in the glass reactor, very similar values of catalytic activity were noticed (Table 5). However, when using the polypropylene reactor, almost 8-fold higher conversion values were reached for the immobilized lipase compared to the free one. The results presented for the free lipase demonstrate that the reactor material had no significant impact on the achieved catalytic parameters. In turn, the data for the immobilized form proved a negative impact of the glass reactor on the catalytic activity because the activity of the immobilized lipase remained at the level of the activity of the free lipase. Changing the reactor to the polypropylene one, while maintaining the other reaction conditions, greatly contributed to the increase in the immobilized CALB activity. This may be one of the arguments for the significant influence of the reactor material on the CALB-octyl-agarose beads.

The issue of the reactor material in the application of the octyl-agarose support is only briefly discussed in the literature. The authors provide information about reactors, often limited to the information: “reactor vial” or “hermetic vial” [46,47]. Our research pointed out a significant impact of the reactor material on the achieved biocatalytic reaction results in an organic solvent. Therefore, the data presented in this paper may provide an argument to the debate on the influence of the reactor materials used in kinetic resolution when applying a reaction medium other than an aqueous one.

#### 2.2.2. Effect of Reaction Medium

As a part of the next step, the reaction medium’s influence on the immobilized CALB’s activity was tested. The reaction was carried out in five different organic solvents using the polypropylene reactor. The lipase was immobilized in a Tris base buffer (pH 9; 100 mM). The reaction was conducted for 48 h. The results are presented in Figure 3. Attention was paid to the logP value and an attempt was made to determine whether there was a relationship between the logP value of the solvent and the conversion. 

In the presented data, the biocatalysts applied in solvents with a higher logP value were characterized by a higher catalytic activity as compared to a reaction medium with a lower logP value. Based on the results, there is a relationship between the logP of the solvent and the enzymatic activity. The conversion values of the reaction increased with the increasing logP values of the solvent. The logP values used in the studies are 1.25, 1.48, 1.98, 1.52, and 1.06 for DCM, DCE, DCP, DIPE, and MTBE, respectively (data received from the suppliers).

The literature has pointed out that a hydrophobic medium promotes a higher catalytic activity, while hydrophilic solvents have a more potential tendency to strip bound water (essential for catalytic activity) from the biocatalyst molecules [48,49]. It should be mentioned that the solubility of the substrate, hydration level, the flexibility of the protein, the stabilization of the charged transition state, and competitive inhibition by solvent are the major factors that influence enzyme activity in organic solvents. The structure and flexibility of the CALB are not greatly influenced by the different organic media. The impact of the solvent molecules depends on the inhibitory effect of the solvent and the solubility of the substrate in the different media. The solvent molecules compete with the substrate for the binding site pocket of the CALB, and work as inhibitors, depending on the strength of the interaction between media molecules and residues in the CALB [50].

Some authors [34] have suggested that the factor limiting the use of agarose (even functionalized agarose) in organic solvents may relate to the fact that water is retained in the support, which may displace the equilibrium of the reaction. Moreover, they paid attention to the possibility of the retention of the hydrophilic substrates or reaction products in the agarose matrix, which may lead to the loss of enzyme activity or mass transfer problems. In our studies, we applied molecular sieves to avoid the problem of water produced as a by-product of the esterification reaction.

#### 2.2.3. Climatic Chamber Stability Tests of CALB

The stability studies of the CALB in a dry form in a climatic chamber were performed under extreme conditions of temperature (65 °C) and humidity (75%). Additionally, the effect of light in the visible spectral range (400–800 nm) on the biocatalyst stability was examined. Moreover, the immobilized lipase was also stored for 7 days in a refrigerator (4 °C). After 7 days of storage, an analysis of the enantioselectivity was performed and the conversion values were determined. The data were compared with the values obtained for the biocatalysts that were not stored (used for the reaction after immobilization and drying). The immobilization of the lipase was carried out in the polypropylene reactor in a Tris base buffer (pH 9; 100 mM). The kinetic resolution was performed by the enantioselective esterification of (*R*,*S*)-flurbiprofen with methanol. The results are shown in Figure 4.

Based on the received data, it was stated that the immobilized CALB is characterized by good stability under the tested conditions (protected from humidity). Furthermore, we noticed that the storage of the biocatalysts in dry form in a climatic chamber and a refrigerator after the immobilization had a positive influence on the catalytic activity. It should be emphasized that humidity during storage was a factor that negatively affected the enzyme activity. The biocatalysts subjected to high humidity (75%) in a climatic chamber presented a lower catalytic activity in an organic medium.

The immobilized enzyme after storage in a climatic chamber for 7 days in a closed polypropylene reactor protected from light and exposed to light was observed to offer approximately 60% higher conversion values as compared to the immobilized CALB not stored in the climatic chamber (c = 39.7 ± 2.0%, c = 40.5 ± 2.3%, and c = 24.7 ± 2.0% after 24 h of reaction, respectively).

However, storing the immobilized biocatalyst in an open polypropylene reactor, protected from light as well as exposed to light, resulted in lowering the CALB activity, which may indicate the negative impact of high humidity. After 24 h of reaction, the conversion of the samples protected and exposed to light was 9.8 ± 1.3% and 9.5 ± 1.3%, respectively. The decrease in activity caused by storing at a high humidity probably relates to the kinetic resolution being carried out in a non-aqueous medium (organic solvent), and the immobilized enzyme being dried before being added to the reaction. It should be noted that samples stored in the refrigerator, protected from light and humidity, offered approximately 60% higher values of conversion than the samples tested without storage (c = 40.5 ± 3.3%, and c = 24.7 ± 2.0%, after 24 h of reaction, respectively).

The results suggest a significant impact of storing the immobilized lipase in a dry form on its activity in organic solvents (enantioselective esterification). The results after storage allow us to conclude about the high stability of the created catalytic systems. The noticed approximately 60% higher conversion values, after the same reaction time, for the immobilized lipase after storage (c = 39.7 ± 2.0% after 24 h) compared to the immobilized lipase without storage (24.7 ± 2.0% after 24 h) is the basis for including this stage to the optimization procedures of the enzymatic activity. The data demonstrate a potential positive effect of the immobilization conditions being used, Tris base buffer (pH 9; 100 mM), on the CALB stability and catalytic activity in kinetic resolution in an organic medium. It is believed that the immobilized lipase (protected from humidity) stored in a climatic chamber, as well as in a refrigerator, maintains its open form.

Our previous papers [17,35] discussed the effect of storage in a climatic chamber and refrigerator on the stability of the CALB immobilized on octyl-agarose regarding its lipolytic activity assessed in an aqueous medium. The results we demonstrate in this paper are consistent with previous ones and confirm the observed phenomenon of an increase in the activity of the immobilized lipase after storage in a climatic chamber and a refrigerator. It should be noted that in the current project, we evaluated lipase activity in the organic reaction medium in the enantioselective esterification reaction—enantioselectivity. Therefore, it can be concluded that the storage stage has a positive impact on the immobilized CALB used in both aqueous and organic reaction media. An important fact is that the stability study of the CALB immobilized on octyl-agarose performed in a climatic chamber and the subsequent application in a dry form in a kinetic resolution in an organic solvent is so far described in the literature to a very limited extent.

As discussed in our previous papers [17,35], it is believed that the observed increase in catalytic activity may result from the elements of the CALB structure—including the lid. Additionally, it is thought that the increase in activity in an organic solvent after storing the tested lipase in an immobilized form at an extremely high temperature could be linked with, among others, changes in the enzyme’s conformation. For comparison, the free CALB was stored in a climatic chamber and a refrigerator for a week. After storage in the climatic chamber, no catalytic activity was documented, which may suggest that the enzyme was denatured. However, after storage in the refrigerator, the catalytic activity parameters remained at a similar level (conversion not higher than 1.5%) as in the procedure without storage.

Based on the results received, it is justified to include the step of storing the CALB immobilized on octyl-agarose in a dry form in the protocol for preparing the biocatalyst for application in reactions carried out in organic solvents. It should be emphasized that the application of optimal immobilization protocols made it possible to obtain catalytic systems with good stability.

### 2.3. CRL-OF—Effect of Immobilization on Enzymatic Activity

The biocatalysts used in the current research, the CRL-OF and CALB, vary in their molecular structure. The former one, unlike the CALB, has a lid that isolates the active site of the enzyme from the external medium, while in the second enzyme, the presence of a lid is still being investigated [35]. It is believed that the various data on catalytic activity received in an organic medium may be the result of the differences in the structure of the tested enzymes. The lipase from *Candida rugosa* (CRL-OF) was immobilized onto octyl-agarose support with the use of phosphate buffer (pH 7 and 100 mM). Then, the immobilized enzyme was used in the kinetic resolution of (*R*,*S*)-flurbiprofen by enantioselective esterification with methanol. The reaction was performed in the polypropylene reactor in organic solvents. The parameters of the catalytic activity: c, ee_p_, and ee_s_ were determined. The results are presented in Table 6.

The results indicate a low catalytic activity of the immobilized CRL-OF in the enantioselective esterification reaction. Additionally, when comparing the data for the immobilized lipase with the data for the free lipase (c = 1.6 ± 0.5%; ee_p_ = 18.1 ± 0.4%; ee_s_ = 0.3 ± 0.1%; in DCP; in the polypropylene reactor; after 24 h of reaction), it is evident that the catalytic activities of both the biocatalyst forms were similar. This may demonstrate only a small impact of the applied immobilization procedures on the increase in lipase activity. This observation could be linked to the structure of the lipase, especially the presence of a lid. 

Despite the low immobilized lipase activity, an attempt to examine the effect of its storage in a refrigerator and a climatic chamber on the enantioselectivity was made. After 7 days of storage in a climatic chamber, no catalytic activity was noticed, which may point to protein denaturation. It should be emphasized that after 7 days of storage in the refrigerator (c = 1.4 ± 0.4%; ee_p_ = 20.3 ± 0.5%; ee_s_ = 0.3 ± 0.1%), similar values of activity as in the case of the immobilized lipase without storage were documented. The data on the lipase activity are characterized by similar trends to those presented and discussed in our previous papers [17,35], where we analyzed the lipolytic activity of the CRL-OF in an aqueous medium.

The effect of the storage of the CRL-OF in a free form in a climatic chamber and a refrigerator on its activity was also tested. After storage in the climatic chamber, we assumed that the denaturation of the catalytic protein had taken place since no catalytic activity was reported in the enantioselective esterification reaction. However, after 7 days of storage in the refrigerator, the activity of the free CRL-OF in the kinetic resolution of (*R*,*S*)-flurbiprofen remained at a comparable level (conversion not higher than 2%) as in the case of the lipase not subjected to the storage stage. Due to the low enzymatic activity obtained for the free and immobilized lipase forms, no further studies of this lipase were conducted.

## 3. Materials and Methods

### 3.1. Materials

Octyl-Sepharose CL-4B (GE Healthcare, Uppsala, Sweden), (*R*,*S*)-flurbiprofen, (*R*)-flurbiprofen, methanol, *n*-heptane, 2-propanol, trifluoroacetic acid, 1,2-dichloropropane (DCP), diisopropyl ether (DIPE), *tert*-butyl methyl ether (MTBE), Bradford reagent, hydrochloric acid, and Tris base were purchased from Sigma-Aldrich (Steinheim, Germany). Dichloromethane (DCM), 1,2-dichloroethane (DCE), citric acid, disodium hydrogen phosphate dihydrate, monosodium hydrogen phosphate monohydrate, and molecular sieves 4 Å were gained from POCH (Gliwice, Poland). Trisodium citrate was from Chempur (Piekary Ślaskie, Poland). The lipase B from *Candida antarctica* (CALB, produced in yeast) was from ChiralVision (Leiden, The Netherlands), and the lipase from *Candida rugosa* (CRL-OF) from Meito Sangyo (Tokyo, Japan). 

### 3.2. Equipment

The amount of the immobilized lipase was estimated using a UV-Vis spectrophotometer U-1800 (Hitachi, Tokyo, Japan). The water used in this investigation was purified by the Milli-Q Water Purification System (Millipore, Bedford, MA, USA). The storage stability of the lipases in the free and immobilized form was tested and studied in a climatic chamber KBF P240 (Tuttlingen, Germany). The buffers were prepared by a SevenMulti pH-meter (Mettler-Toledo, Schwerzenbach, Switzerland). The octyl-Sepharose CL-4B support was prepared by centrifuge Eppendorf MiniSpin Plus (Hamburg, Germany) and mixer vortex Velp Scientifica ZX4 (Usmate, Italy). The incubation of the samples was performed in Thermomixer comfort (Eppendorf AG, Hamburg, Germany). The analysis of the kinetic resolution of (*R*,*S*)-flurbiprofen was conducted with the application of HPLC. The Shimadzu HPLC system (Kyoto, Japan) is composed of a pump (LC-20 AD), a UV–VIS detector (SPD-20A), a degasser (DGU-20A_5R_), an autosampler (SIL-20AC_HT_), and a column oven (CTO-10AS_VP_). As a chiral selector, a Lux Cellulose-3 (LC-3) (4.6 mm × 250 mm) column with cellulose tris(4-methylbenzoate) and pre-column (Guard Cartridge System, KJO-4282 model) was applied. The column had a 5 μm particle size. The most appropriate chromatographic conditions for (*R*)- and (*S*)-flurbiprofen and their esters were established with *n*-heptane/2-propanol/trifluoroacetic acid (98.1/1.9/0.2, *v*/*v*/*v*) mobile phase at a flow rate of 1 mL/min. The UV detection wavelength was set at 254 nm. The temperature of the chromatographic process was 15 °C.

The enantiomeric excesses of the substrate (ee_s_) and the product (ee_p_), as well as the conversion (c), were calculated using the following equations [18].

The ee_s_ and ee_p_ values were expressed as follows: $$\left(\%\right){ee}_{s}=\frac{\left|R_{s}-S_{s}\right|}{R_{s}+S_{s}}\times 100$$
$${\left(\%\right)ee}_{p}=\frac{\left|{R_{p}-S}_{p}\right|}{R_{p}+S_{p}}\times 100$$

R_s_, S_s_—the enantiomers of the substrate (*R*,*S*-flurbiprofen); represent the peak areas of the *R*- and *S*-enantiomers, respectively.

R_p_, S_p_—the enantiomers of the product (methyl ester of (*R*,*S*)-flurbiprofen); represent the peak areas of the *R*- and *S*-enantiomers, respectively.

The conversion (c) was expressed as follows:$$\;\left(\%\right)c=\frac{{ee}_{s}}{{ee}_{s}+{ee}_{p}}\times 100$$

### 3.3. Methods

#### 3.3.1. Octyl-Agarose Preparation Technique

The octyl-agarose bead suspension (110 μL) was inserted into a polypropylene or a glass tube. In the next step, 1 mL of filtered water was added into the tube with support suspension, and the mixture was stirred by applying a vortex for 3 min, followed by centrifugation for 15 min at 9000 rpm. Afterwards, the support, after separation from the supernatant, was weighed (50 mg).

#### 3.3.2. Immobilization of CALB onto Octyl-Agarose

The immobilization method was carried out in our laboratory with slight changes [17,35]. An amount of 10.0 mg of the CALB was placed in an Eppendorf tube (2.0 mL) and suspended in 1.0 mL of an appropriate buffer. The sample stayed for 15 min at room temperature. After this time, the CALB suspension was mixed and then placed into the polypropylene or glass vial (2.0 mL) containing 50 mg of the prepared octyl-agarose support. The samples were mixed for 5 min and then maintained at 4 °C for 14 h. Finally, the supernatant was collected, and the supports with the immobilized lipase were air-dried for 48 h. The procedures were performed in triplicate. For immobilization, the following buffers were used: citric buffer (pH 4; 100 mM), phosphate buffer (pH 7; 100 mM), and Tris base buffer (pH 9; 50 mM, 100 mM, 300 mM, and 500 mM).

#### 3.3.3. Immobilization of CRL-OF onto Octyl-Agarose

The immobilization method was carried out in our laboratory with slight change [17,35]. An amount of 10.0 mg of the CRL-OF was placed in an Eppendorf tube (2.0 mL) with 1.0 mL of an appropriate buffer (phosphate buffer—pH 7; 100 mM, and Tris base buffer—pH 9; 100 mM). The sample was kept at room temperature for 15 min. After this time, the CRL-OF suspension was mixed and then placed into the polypropylene vial (2.0 mL) containing 50 mg of the prepared octyl-agarose support. The samples were mixed for 5 min and then kept at 4 °C for 14 h. Finally, the supernatant was collected, and the supports with the immobilized lipase were air-dried for 48 h. The procedures were performed in triplicate.

#### 3.3.4. Determination of the Amount of Immobilized CALB

The amount of the immobilized CALB was determined using the Bradford method [17,35,51] with a few modifications. The study was carried out using the UV-Vis spectrophotometric method (λ = 595.0 nm), measuring the absorbance of the free lipase remaining in the suspension after the immobilization process (concentration range: 1.0–10.0 mg/mL). The measurement was made in triplicate. The amount of the CALB immobilized onto the octyl-agarose beads was calculated with a calibration curve equation (R^2^ = 0.999 ± 0.004). The result was the three-sample mean. The lipase loading (mg/g support) was determined based on the received data. The process was repeated for buffers with various pHs (4, 7, and 9); 100 mM.

#### 3.3.5. Enantioselectivity—Kinetic Resolution of (*R*,*S*)-Flurbiprofen

The kinetic resolution of (*R*,*S*)-flurbiprofen was performed according to the method described in the literature [18,19], including some changes. The (*R*,*S*)-flurbiprofen (4.8 mg, 0.02 mM), reaction medium (DCM/DCE/DCP/DIPE/MTBE; 700 μL), methanol (2.44 μL, 0.06 mM), and molecular sieve 4 Å (reactions with free form were carried out without the use of molecular sieves) were added to the polypropylene or glass vial containing the free enzyme (CALB or CRL-OF) or the octyl-agarose support with the immobilized lipase (CALB or CRL-OF). The sample vials were closed and secured with thermal insulation tape and then incubated in Thermomixer with mixing (37 °C, 550 rpm). The reaction time was 18, 22, 24, and 48 h. Afterwards, the samples (50 μL) were collected and dried at room temperature and then dissolved (2-propanol; 0.9 mL), and after filtration (0.45 μm), injected (5 μL) into the HPLC column. Analyses were performed in triplicate.

#### 3.3.6. Stability Tests of CALB and CRL-OF in Dry Form—Climatic Chamber and Refrigerator

The stability test of the immobilized CALB and CRL-OF was performed according to the literature method [17,35]. After immobilization in the Tris base buffer (pH 9, 100 mM), the supernatant was collected and the supports with the immobilized lipase were air-dried for 48 h. Then, the octyl-agarose beads with the immobilized enzymes were stored in polypropylene vials in a climatic chamber KBF P240 or refrigerator (4 °C). In the climatic chamber, the temperature was maintained at 65 °C, the humidity was 75%, and the visible spectral range was 400–800 nm. The immobilized lipases were stored for 7 days. Then, the enantioselectivity of the immobilized lipases was evaluated according to Section 3.3.5 (the reaction time was 24 h). The storage of the octyl-agarose support with the immobilized lipase was performed in combinations as follows:Climatic chamber—65 °C (without humidity and light-Vis)Climatic chamber—65 °C with light-Vis (without humidity)Climatic chamber—65 °C with humidity 75% (without light-Vis)Climatic chamber—65 °C with humidity 75% and light-VisRefrigerator—4 °C (without humidity and light-Vis)

## 4. Conclusions

A new approach of biocatalysis carried out in an organic solvent with the use of a CALB-octyl-agarose support was based on the application of a polypropylene reactor, the optimization of the buffer for the immobilization, and the use of a drying procedure. The paper presents new protocols, not described in the literature, for obtaining immobilized (on octyl-agarose) lipase B from *Candida antarctica* for potential use in receiving pure enantiomers from the group of 2-arylpropionic acid derivatives by enantioselective esterification. The proposed immobilization conditions (Tris base buffer with pH 9 and 100 mM), the drying step of the biocatalyst, and the use of polypropylene reactor allowed for reaching a high enantioselectivity (*R*-flurbiprofen methyl ester with ee_p_ = 89.6 ± 2.0%) of the immobilized CALB. The possibility of controlling the catalytic activity of the lipase by selecting the parameters of the buffer (pH, concentration) for immobilization was demonstrated. Additionally, a strong relationship between the activity of the immobilized CALB (on octyl-agarose) and the reactor material used in the reaction carried out in an organic medium has been proven. In the application of a reactor prepared from polypropylene, the activity (conversion) was over 9-fold higher when compared to the glass reactor. Stability tests conducted in a climatic chamber and a refrigerator indicated the good stability of the created catalytic system. Approximately 60% higher conversion values were received after the storage (protected from humidity) of the immobilized CALB, than the lipase which was not stored. The use of a polypropylene reactor at the stage of the immobilization and reaction performed allowed us to obtain approximately 30-fold higher conversion values compared to the free lipase. On the other hand, regarding the CRL-OF, the low enantioselectivity of the immobilized form was shown. The new approach described in this paper draws attention to the possibility of using polymers as materials for preparing reactors to conduct the reaction in an organic solvent, catalyzed by enzymes immobilized on octyl-agarose. 

## Figures and Tables

**Figure 1 ijms-25-05084-f001:**
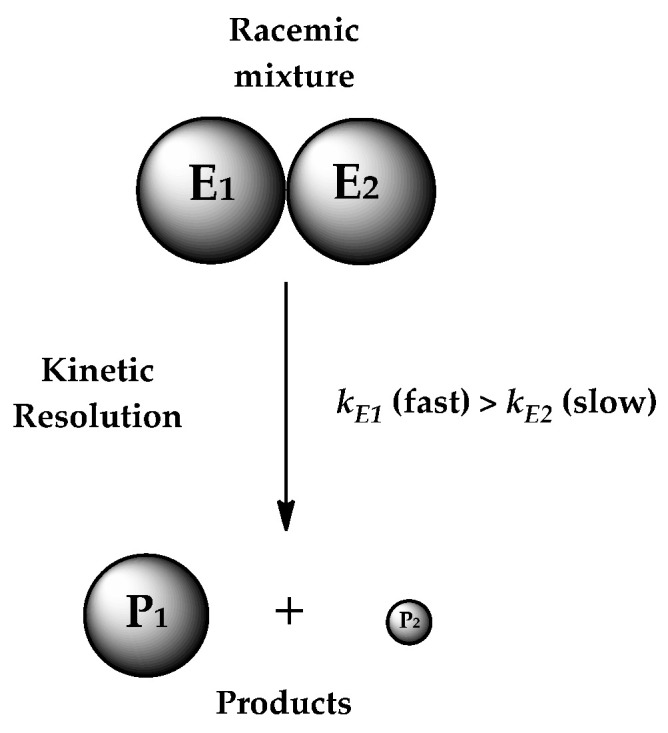
The simplified reaction of the kinetic resolution of a racemic mixture [11]. E1 and E2 refer to the enantiomers of the racemic mixture, *k* refers to the reaction rate constant, and P1 and P2 refer to the products of the kinetic resolution of the racemic mixture of the enantiomers E1 and E2, respectively.

**Figure 2 ijms-25-05084-f002:**
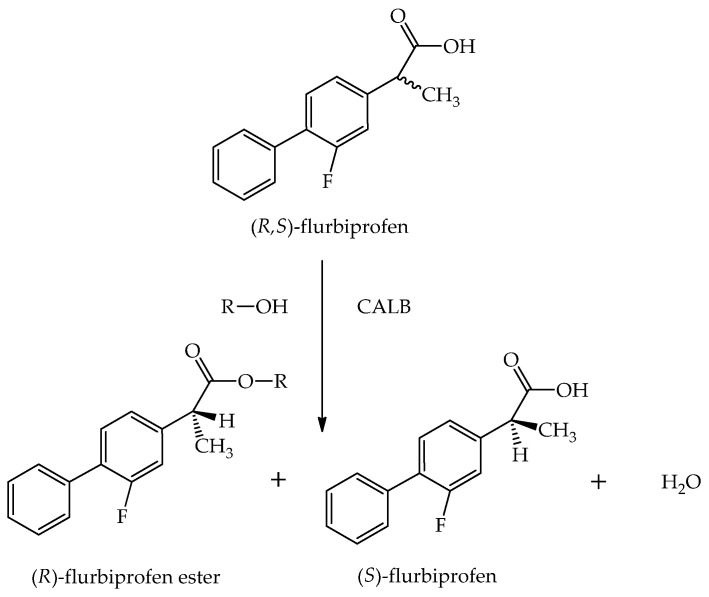
The kinetic resolution of (*R*,*S*)-flurbiprofen via enzymatic esterification, catalyzed by the lipase B from *Candida antarctica* (CALB) [18]. (*R*,*S*)-flurbiprofen is the donor of the acyl group, whereas the alcohol is characterized as an acyl acceptor.

**Figure 3 ijms-25-05084-f003:**
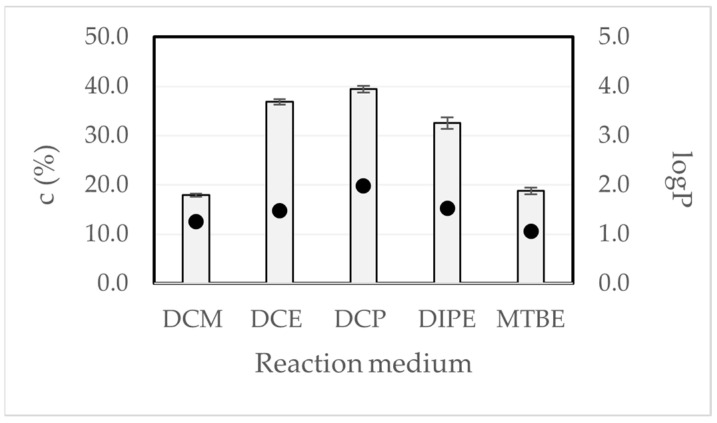
Effect of the reaction medium on the reaction conversion values. Reaction conditions: racemic flurbiprofen (4.8 mg, 0.02 mM), methanol (2.44 μL, 0.06 mM), immobilized CALB (50 mg of support), medium (dichloromethane—DCM, 1,2-dichloroethane—DCE, 1,2-dichlopropane—DCP, diisopropyl ether—DIPE, and *tert*-butyl methyl ether—MTBE) (700 μL), molecular sieve 4 Å, reaction temp 37 °C, and shaking at 600 rpm; polypropylene reactor; c —conversion. Data are presented as means ± standard deviations of the three analyses (*n* = 3). The error bars represent the standard deviations of the mean. The dots refer to the logP value of the reaction medium.

**Figure 4 ijms-25-05084-f004:**
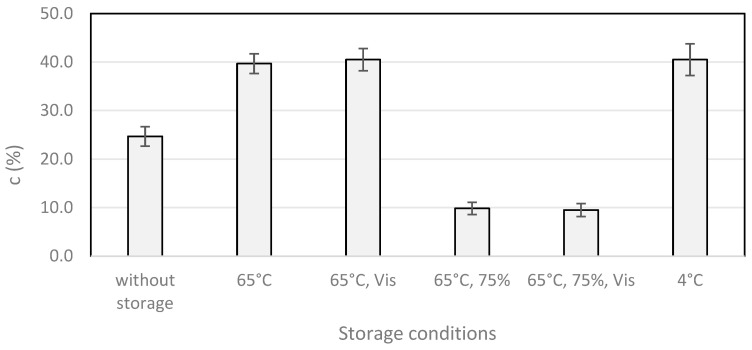
Effect of the storage conditions in the climatic chamber and refrigerator on the storage stability of the immobilized CALB in dry form. Reaction conditions: racemic flurbiprofen (4.8 mg, 0.02 mM), methanol (2.44 μL, 0.06 mM), immobilized CALB (50 mg of support), medium (DCP) (700 μL), molecular sieve 4 Å, reaction temp 37 °C, and shaking at 600 rpm; reaction time—24 h; polypropylene reactor; c —conversion. Data are presented as means ± standard deviations of the three analyses (*n* = 3). The error bars represent the standard deviations of the mean. The “Vis” means the presence of light in the visible spectral range (400–800 nm), and the 75% means the value of humidity in the climatic chamber.

**Table 1 ijms-25-05084-t001:** The effect of the buffer pH on the values of the conversion, enantiomeric excess, and lipase loading.

Lipase	Reactor Material	Reaction Medium	Lipase Loading (mg/g Support)	Reaction Time	Immobilization Conditions	ee_s_ (%)	ee_p_ (%)	c (%)
CALB	polypropylene	DCP	pH 4 64.8 ± 0.9	24 h	pH 4; 100 mM	3.9 ± 0.2	63.0 ± 0.8	5.8 ± 0.1
					pH 7; 100 mM	4.9 ± 0.3	74.4 ± 2.4	6.2 ± 0.2
			pH 7 42.1 ± 1.4		pH 9; 100 mM	29.6 ± 0.6	89.6 ± 2.0	24.9 ± 0.1
				48 h	pH 4; 100 mM	11.3 ± 0.8	63.4 ± 1.8	15.1 ± 0.5
			pH 9 30.5 ± 0.6		pH 7; 100 mM	13.7 ± 0.4	73.1 ± 3.0	15.8 ± 0.2
					pH 9; 100 mM	51.1 ± 1.2	80.5 ± 0.4	38.8 ± 0.5

Reaction conditions: racemic flurbiprofen (4.8 mg, 0.02 mM), methanol (2.44 μL, 0.06 mM), immobilized CALB (50 mg of support), medium (DCP) (700 μL), molecular sieve 4 Å, reaction temp 37 °C, and shaking at 600 rpm; polypropylene reactor; c—conversion, ee_s_—the enantiomeric excess of the substrate, and ee_p—_the enantiomeric excess of the product. Data are presented as means ± standard deviations of the three analyses (*n* = 3).

**Table 2 ijms-25-05084-t002:** Effect of the buffer concentration on the values of the conversion and enantiomeric excess.

Lipase	Reactor Material	Reaction Medium	Immobilization Conditions	Reaction Time	ee_s_ (%)	ee_p_ (%)	c (%)
CALB	polypropylene	DCP	pH 9; 50 mM	24 h	17.7 ± 0.8	82.5 ± 0.8	17.7 ± 0.5
				48 h	43.8 ± 1.2	80.2 ± 0.9	35.3 ± 0.4
			pH 9; 100 mM	24 h	29.6 ± 0.6	89.6 ± 2.0	24.9 ± 0.1
				48 h	51.1 ± 1.2	80.5 ± 0.4	38.8 ± 0.5
			pH 9; 300 mM	24 h	32.8 ± 1.9	75.7 ± 1.5	30.2 ± 0.9
				48 h	36.6 ± 0.7	63.9 ± 1.5	36.4 ± 0.1
			pH 9; 500 mM	24 h	4.8 ± 0.3	43.6 ± 1.7	9.9 ± 0.1
				48 h	6.8 ± 0.5	43.1 ± 1.5	13.7 ± 0.5

Reaction conditions: racemic flurbiprofen (4.8 mg, 0.02 mM), methanol (2.44 μL, 0.06 mM), immobilized CALB (50 mg of support), medium (DCP) (700 μL), molecular sieve 4 Å, reaction temp 37 °C, and shaking at 600 rpm; polypropylene reactor; c—conversion, ee_s_—the enantiomeric excess of the substrate, and ee_p—_the enantiomeric excess of the product. Data are presented as means ± standard deviations of the three analyses (*n* = 3).

**Table 3 ijms-25-05084-t003:** The values of the conversion and enantiomeric excess of the free CALB.

Lipase	Reactor Material	Reaction Medium	Reaction Time	ee_s_ (%)	ee_p_ (%)	c (%)
CALB (free)	polypropylene	DCP	24 h	1.0 ± 0.1	99.9 ± 0.0	1.0 ± 0.1
			48 h	1.2 ± 0.2	99.9 ± 0.0	1.2 ± 0.1

Reaction conditions: racemic flurbiprofen (4.8 mg, 0.02 mM), methanol (2.44 μL, 0.06 mM), free CALB (10 mg), medium (DCP) (700 μL), reaction temp 37 °C, and shaking at 600 rpm; polypropylene reactor; c —conversion, ee_s_—the enantiomeric excess of the substrate, ee_p—_the enantiomeric excess of the product. Data are presented as means ± standard deviations of the three analyses (*n* = 3).

**Table 4 ijms-25-05084-t004:** The effect of the reactor material on the value of the conversion and enantiomeric excess.

Lipase	Reactor Material	Reaction Medium	Reaction Time	Immobilization Conditions	ee_s_ (%)	ee_p_ (%)	c (%)
CALB	glass	DCM	24 h	pH 7; 100 mM	1.4 ± 0.1	99.9 ± 0.0	1.4 ± 0.1
	polypropylene	DCM	18 h	pH 7; 100 mM	12.6 ± 0.5	90.7 ± 1.0	12.2 ± 0.3
	glass	DCP	22 h	pH 7; 100 mM	0.8 ± 0.3	39.6 ± 0.7	2.0 ± 0.6
	polypropylene	DCP	24 h	pH 7; 100 mM	4.9 ± 0.3	74.4 ± 2.4	6.2 ± 0.2
	glass	DCP	24 h	pH 9; 100 mM	0.9 ± 0.2	32.8 ± 2.1	2.6 ± 0.4
	polypropylene	DCP	24 h	pH 9; 100 mM	29.6 ± 0.6	89.6 ± 2.0	24.9 ± 0.1

Reaction conditions: racemic flurbiprofen (4.8 mg, 0.02 mM), methanol (2.44 μL, 0.06 mM), immobilized CALB (50 mg of support), medium (DCM, and DCP) (700 μL), molecular sieve 4 Å, reaction temp 37 °C, and shaking at 600 rpm; polypropylene or glass reactor; c—conversion, ee_s_—the enantiomeric excess of the substrate, ee_p_—the enantiomeric excess of the product. Data are presented as means ± standard deviations of the three analyses (*n* = 3).

**Table 5 ijms-25-05084-t005:** The values of the conversion and enantiomeric excess of the free CALB in different reactor materials.

Lipase	Reactor Material	Reaction Medium	Reaction Time	ee_s_ (%)	ee_p_ (%)	c (%)
CALB (free)	glass	DCM	24 h	1.5 ± 0.2	99.9 ± 0.0	1.5 ± 0.1
	polypropylene	DCM	24 h	1.6 ± 0.2	99.9 ± 0.0	1.6 ± 0.1

Reaction conditions: racemic flurbiprofen (4.8 mg, 0.02 mM), methanol (2.44 μL, 0.06 mM), free CALB (10 mg), medium (DCM) (700 μL), reaction temp 37 °C, and shaking at 600 rpm; polypropylene or glass reactor; c—conversion, ee_s_—the enantiomeric excess of the substrate, ee_p_—the enantiomeric excess of the product. Data are presented as means ± standard deviations of the three analyses (*n* = 3).

**Table 6 ijms-25-05084-t006:** The values of the conversion and enantiomeric excess of the immobilized CRL-OF in different reaction media.

Lipase	Reactor Material	Reaction Medium	Reaction Time	ee_s_ (%)	ee_p_ (%)	c (%)
Immobilized CRL-OF	polypropylene	DCM	24 h	-	-	-
		DCE		0.7 ± 0.1	33.6 ± 0.5	2.0 ± 0.3
		DCP		0.4 ± 0.1	22.3 ± 0.6	1.8 ± 0.4
		DIPE		0.8 ± 0.1	39.6 ± 0.6	2.0 ± 0.2
		MTBE		-	-	-

Reaction conditions: racemic flurbiprofen (4.8 mg, 0.02 mM), methanol (2.44 μL, 0.06 mM), immobilized CRL-OF (50 mg of support), medium (dichloromethane—DCM, 1,2-dichloroethane—DCE, 1,2-dichlopropane—DCP, diisopropyl ether—DIPE, *tert*-butyl methyl ether—MTBE) (700 μL), molecular sieve 4 Å, reaction temp 37 °C, and shaking at 600 rpm; polypropylene reactor; c—conversion, ee_s_—the enantiomeric excess of the substrate, ee_p_—the enantiomeric excess of the product. Data are presented as means ± standard deviations of the three analyses (*n* = 3).

## Data Availability

Data are contained within the article.

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
