# Peer review of "A New Approach in Lipase-Octyl-Agarose Biocatalysis of 2-Arylpropionic Acid Derivatives"

_ijms, 2024, doi:10.3390/ijms25105084_

Round 1
Reviewer 1 Report
Comments and Suggestions for Authors
The article “A new approach in lipase-octyl-agarose biocatalysis of 2-arylpropionic acid derivatives” is devoted to a current topic, has high applied significance, and is well structured.
This article described a new approach to biocatalysis performed in an organic solvent with the use of CALB-octyl-agarose support including the application of a polypropylene reactor, appropriate buffer for immobilization (Trizma Base buffer with pH 9 and ionic strength of 100 mM), drying step, and next storage of immobilized lipases in a climatic chamber or refrigerator.
In this article, the authors perfectly demonstrated the dynamics of the development of their research, indicating in detail what they had done previously within the framework of this topic.
In general, the article makes a very favorable impression, but I would like to recommend that the authors add mechanisms that determine the observed effects of buffer pH, buffer ionic strength, reactor material, reaction medium on lipase activity. They can be given on the basis of literature data for other lipases, or authors can make their own calculations using molecular dynamics methods.
Technical Note:
Line 431: instead of “active center”, “active site” would be more successful.
Author Response
Responses to Reviewer 1
- English language fine. No issues detected
Thank you for positively assessing the paper in terms of language.
- Is the research design appropriate? Must be improved
Thank you for your assessment. The research design has been improved.
- Are the results clearly presented? Can be improved
Thank you for your assessment. The results presentation has been improved.
- Are the conclusions supported by the results? Can be improved
Thank you for your assessment. The conclusions supported by the results have been improved.
- The article “A new approach in lipase-octyl-agarose biocatalysis of 2-arylpropionic acid derivatives” is devoted to a current topic, has high applied significance, and is well structured.
This article described a new approach to biocatalysis performed in an organic solvent with the use of CALB-octyl-agarose support including the application of a polypropylene reactor, appropriate buffer for immobilization (Trizma Base buffer with pH 9 and ionic strength of 100 mM), drying step, and next storage of immobilized lipases in a climatic chamber or refrigerator.
In this article, the authors perfectly demonstrated the dynamics of the development of their research, indicating in detail what they had done previously within the framework of this topic.
We would like to thank you for your comment.
- In general, the article makes a very favorable impression, but I would like to recommend that the authors add mechanisms that determine the observed effects of buffer pH, buffer ionic strength, reactor material, reaction medium on lipase activity. They can be given on the basis of literature data for other lipases, or authors can make their own calculations using molecular dynamics methods.
Thank you for your comments. Appropriate information, following the reviewer's suggestion, was introduced into the manuscript. The changes introduced are marked in green in the manuscript.
Line 227-241
Line 290-298
Line 338-341
Line 393-401
- Technical Note:
Line 431: instead of “active center”, “active site” would be more successful.
Thank you for your comments. Appropriate correction, following the reviewer's suggestion, was introduced into the manuscript.
Line 492
The former one, unlike the CALB, has a lid that isolates the active site of the enzyme from the external medium, while in the second enzyme, the presence of a lid is still being investigated [35].

Reviewer 2 Report
Comments and Suggestions for Authors
General: This paper describes a new procedure for the immobilization of Candida antartica lipase B (CALB) on octyl Sepharose. The performance of the immobilized enzyme in organic solvent (1,2-dichloropropane) was evaluated by measuring the kinetic resolution of (R,S)‑flurbiprofen by enantioselective esterification with methanol. New aspects of the immobilization procedure involved the application of a polypropylene reactor, an optimization of the immobilization conditions, a drying step, and storage of the immobilized lipase in a climatic chamber or refrigerator. Next, the effects of reactor material (glass, polypropylene) and reaction medium (organic solvent) were addressed. Storage stability tests (climate chamber, refrigerator, effect of visible light, humidity) revealed that storage of the immobilized CALB in dry form is essential for the high activity and enantioselectivity of the enzyme. Finally, the performance of Candida rugosa lipase, immobilized in a similar way was tested. This indicated that the procedure developed for CALB is not suitable for this lid-containing lipase.
The paper contains extensive information about the used procedures. That is good, however, there is quite some repetition of certain explanations. In my view, the text can be shortened a lot with clear statements about the impact and applicability of the new findings. The molecular explanations about the possible role of the lid remain vague, and as it stands now, the paper seems more suited for a biotechnology or biocatalysis oriented journal.
Comments
39 Keywords: discriminate between storage stability and operational stability
64 Fig. 1 is not fully correct. One of the enantiomers of the racemic mixture is faster converted. This can be indicated by the size of the circles. Furthermore, I see no products.
147 Please note that the ionic strength values of the 100 mM citrate, 100 mM phosphate and 100 mM Tris buffers are not the same. They depend on the concentration and charge of all ions present in the buffer, and thus on the pH and pKa values. See also remark line 200. It cannot be excluded that the pH results may also be partially explained by the differences in ionic strength (or type of ions in the buffer).
151 products or product? See also line 158
154 This table does not report the value of catalytic activity.
154 Do these data say something about the selectivity value (s), i.e. the ratio between kR and kS? The s value would give an indication about the theoretical maximum of conversion and the maximum ee value of the product.
154 Under the best conditions, the ee value of the product drops over time, while the total conversion and ee value of the substrate increase. This suggests that the yield of the S-enantiomer of the substrate becomes lower and lower. What does this mean for the scale up of the reaction?
177 Mention enzyme loading in Table 1?
178 What kind of steric hindrance is meant here?
200 These are buffer concentrations, not ionic strength values. I assume that the Tris buffers were made by titration with HCl, so the concentration of chloride ions needs also to be taken into account.
206 This table does not report the value of catalytic activity.
222 In hydrophobic interaction chromatography, the interaction of the enzyme with the agarose support usually increases with temperature and with high salt conditions. What were the loadings of the enzyme in the pH 9 buffers? In other words, were the conversion values dependent on the amount of loaded enzyme? And would that differ from the situation when immobilizing at pH 4 or pH 7?
316 Relation with log P values should be illustrated in Fig. 3
356 Fig. 4, for clarity, explain the meaning of Vis and 75% also in the legend.
395 Is there a way to test this believing?
424 Remove first part of sentence.
429 To compare the catalytic activities of these biocatalysts, CRL-OF was immobilized….
432 The results with CRL-OF are rather preliminary. Did you also test immobilization of CRL-OF in Tris pH 9? Serendipity cannot be excluded…..
Comments on the Quality of English LanguageMinor editing of the English language required.
Author Response
Responses to Reviewer 2
- Minor editing of English language required.
Thank you for your comment. Editing of the English language has been performed.
- Are the results clearly presented? Must be improved
Thank you for your assessment. The results presentation has been improved.
- Are the conclusions supported by the results? Must be improved
Thank you for your assessment. The conclusions supported by the results have been improved.
- General: This paper describes a new procedure for the immobilization of Candida antartica lipase B (CALB) on octyl Sepharose. The performance of the immobilized enzyme in organic solvent (1,2-dichloropropane) was evaluated by measuring the kinetic resolution of (R,S)‑flurbiprofen by enantioselective esterification with methanol. New aspects of the immobilization procedure involved the application of a polypropylene reactor, an optimization of the immobilization conditions, a drying step, and storage of the immobilized lipase in a climatic chamber or refrigerator. Next, the effects of reactor material (glass, polypropylene) and reaction medium (organic solvent) were addressed. Storage stability tests (climate chamber, refrigerator, effect of visible light, humidity) revealed that storage of the immobilized CALB in dry form is essential for the high activity and enantioselectivity of the enzyme. Finally, the performance of Candida rugosa lipase, immobilized in a similar way was tested. This indicated that the procedure developed for CALB is not suitable for this lid-containing lipase.
We would like to thank you for your comment.
- The paper contains extensive information about the used procedures. That is good, however, there is quite some repetition of certain explanations. In my view, the text can be shortened a lot with clear statements about the impact and applicability of the new findings. The molecular explanations about the possible role of the lid remain vague, and as it stands now, the paper seems more suited for a biotechnology or biocatalysis oriented journal.
We would like to thank you for your comment. The text was modified according to the reviewer's suggestion. The molecular explanations about the possible role of the lid have been introduced to the manuscript. The changes added are marked in green in the manuscript.
Line 59-69
Line 87-89
The influence of lipase structure was also described in our previous publication [35].
Siódmiak, T.; Siódmiak, J.; Mastalerz, R.; Kocot, N.; DulÄ™ba, J.; Haraldsson, G.G.; WÄ…tróbska-Åšwietlikowska, D.; MarszaÅ‚Å‚, M.P. Climatic Chamber Stability Tests of Lipase-Catalytic Octyl-Sepharose Systems. Catalysts 2023, 13, doi:10.3390/catal13030501.
“Lipase B from Candida antarctica belongs to the α/β-hydrolases family, with a catalytic
triad containing Ser-Asp-His. CALB possesses two mobile α -helices (α5 and α10), surrounding the active center, which can act as a lipase lid and contributes to the ability of the enzyme to interact with many different substrates. It should be mentioned that the mechanism of the catalytic activity of the CALB remains the subject of research, to answer the issue of whether CALB catalyzes its reactions by interfacial activation [16,19]. The optimum pH of the reaction medium for CALB is 7.4. It is worth observing that the enzyme activity decreases in the medium with pH below 6 and above 8. This phenomenon is probably related to the ionization state of the amino acid residues of Asp 187 and His 224 from the catalytic triad [20]. The isoelectric point of CALB is at pH 6 [21]. The described lipase is also characterized by high enantioselectivity, which is reflected in its widespread use in the pharmaceutical industry for the preparation of, e.g., pure drug enantiomers [22–25].
The lipase from Candida rugosa is a protein with a molecular weight of 60 kDa, belonging
to, similarly to CALB, the α/β-hydrolases family. The CRL is characterized by high catalytic activity, low costs, and diverse substrate specificity. Thanks to the polypeptide chain, the “lid”, in the presence of a hydrophobic surface, CRL undergoes interfacial activation, which allows the hydrolysis of poorly soluble substrates in water (oils and fats), unlike standard esterases [26–28]. Importantly, yeast Candia rugosa is especially powerful in secreting a subset of five lipase isoforms (identical in 77%), which are characterized by the ability to hydrolyze lipids and esters of cholesterol [29–31]. The CRL specificity is conditioned by the molecular properties of the enzyme, substrate structure, and factors affecting enzyme–substrate interactions [30].”
6) 39 Keywords: discriminate between storage stability and operational stability
Thank you for your comment. Changes in keywords have been made: storage stability
The changes added are marked in green in the manuscript.
7) 64 Fig. 1 is not fully correct. One of the enantiomers of the racemic mixture is faster converted. This can be indicated by the size of the circles. Furthermore, I see no products.
Thank you for your comment. Appropriate changes, as suggested by the reviewer, have been introduced.
Line 78
8) 147 Please note that the ionic strength values of the 100 mM citrate, 100 mM phosphate and 100 mM Tris buffers are not the same. They depend on the concentration and charge of all ions present in the buffer, and thus on the pH and pKa values. See also remark line 200. It cannot be excluded that the pH results may also be partially explained by the differences in ionic strength (or type of ions in the buffer).
Thank you for your comment. Appropriate changes, as suggested by the reviewer, have been introduced.
Line 227
The analysis was based on pH values and buffer concentrations, but it should be noted that the ionic strength and the nature of buffers (chemical composition) used in the procedure may also impact the enzyme activity.
Line 243
The CALB immobilization with the application of buffers at pH 9 and different concentrations (50 mM, 100 mM, 300 mM, 500 mM) were investigated.
9) 151 products or product? See also line 158
Thank you for your comment. Appropriate changes, as suggested by the reviewer, have been introduced.
eep - enantiomeric excess of the product
10) 154 This table does not report the value of catalytic activity.
Thank you for your comment. Appropriate changes, as suggested by the reviewer, have been introduced. The changes added are marked in green in the manuscript.
Line 178
Table 1. The effect of buffer pH on the values of the conversion, enantiomeric excess, and lipase loading.
11) 154 Do these data say something about the selectivity value (s), i.e. the ratio between kR and kS? The s value would give an indication about the theoretical maximum of conversion and the maximum ee value of the product.
Thank you for your comment and question.
Appropriate formulas and explanations have been introduced into the manuscript. Line 576
The enantiomeric excesses of the substrate (ees) and the product (eep) as well as the conversion (C) were calculated using of the following equations [18].
The ees and eep values were expressed as:
Rs, Ss – enantiomers of the substrate (R,S-flurbiprofen); represent the peak areas of the R- and S-enantiomers, respectively
Rp, Sp – enantiomers of the product (methyl ester of (R,S)-flurbiprofen); represent the peak areas of the R- and S-enantiomers, respectively
The conversion (C):
Using the determined values of conversion and enantiomeric excess, the enantioselectivity parameter (E) can be calculated. Enantioselectivity (or enantiomeric ratio, E) is a parameter used to describe the enantioselectivity of a reaction.
In this work, we presented the parameters: C - conversion, ees - enantiomeric excess of the substrate, eep - enantiomeric excess of the product
We performed enzymatic kinetic resolution (theoretically up to 50% conversion)).
12) 154 Under the best conditions, the ee value of the product drops over time, while the total conversion and ee value of the substrate increase. This suggests that the yield of the S-enantiomer of the substrate becomes lower and lower. What does this mean for the scale up of the reaction?
Thank you for your comment and question.
Enzymatic kinetic resolution is one of the methods for obtaining enantiomerically pure compounds. The resolution mechanism is based on the assumption that individual enantiomers have different reaction rates with the biocatalyst. Therefore, one enantiomer is catalyzed at a much higher rate than the other.
Kinetic resolution takes place when the reaction rate constants of the (R)- and (S)- enantiomers of the racemate are different (kR≠kS) and the reaction stops between 0-100% conversion. In the theoretical assumption of kinetic resolution, one of the enantiomers reacts much faster than the other.
In the case of the reaction's scale-up, dynamic kinetic resolution should be applied.
Unlike kinetic resolution (KR), in dynamic kinetic resolution (DKR) the substrates are subjected to constant racemization during resolution, and therefore an equilibrium is created between the (R)- and (S)-enantiomers, which allows for the reaction of all substrates and obtaining only one enantiomer (in our case - the (R)-product: R-flurbiprofen methyl ester) with 100% theoretical efficiency.
13) 177 Mention enzyme loading in Table 1?
Thank you for your comment. Appropriate changes, as suggested by the reviewer, have been introduced. Line 178
14) 178 What kind of steric hindrance is meant here?
Thank you for your comment. Appropriate changes have been introduced.
Line 202-208
The lowest amount of immobilized lipase was received in the pH 9 buffer (lipase loading at pH 9 - 30.5±0.6 mg/g support), which may result in improvement of the substrate’s availability for the lipase during catalysis, as well as probably affecting the enzyme aggregation limitation [18]. Lipase loading in the pH 4 buffer was 64.8±0.9 mg/g support and at the pH 7 - 42.1±1.4 mg/g support. The high loading of the support with biocatalysts causes enzyme crowding on the support surface [38].
15) 200 These are buffer concentrations, not ionic strength values. I assume that the Tris buffers were made by titration with HCl, so the concentration of chloride ions needs also to be taken into account.
Thank you for your comment. Appropriate changes have been introduced.
Line 242-244
2.1.2. Effect of Buffer Concentration
The CALB immobilization with the application of buffers at pH 9 and different concentrations (50 mM, 100 mM, 300 mM, 500 mM) were investigated.
16) 206 This table does not report the value of catalytic activity.
Thank you for your comment. Appropriate changes, as suggested by the reviewer, have been introduced.
Line 249
Table 2. Effect of buffer concentration on the values of the conversion and enantiomeric excess.
17) 222 In hydrophobic interaction chromatography, the interaction of the enzyme with the agarose support usually increases with temperature and with high salt conditions. What were the loadings of the enzyme in the pH 9 buffers? In other words, were the conversion values dependent on the amount of loaded enzyme? And would that differ from the situation when immobilizing at pH 4 or pH 7?
Thank you for your comments and questions.
Line 202-208
The lowest amount of immobilized lipase was received in the pH 9 buffer (lipase loading at pH 9 - 30.5±0.6 mg/g support), which may result in improvement of the substrate’s availability for the lipase during catalysis, as well as probably affecting the enzyme aggregation limitation [18]. Lipase loading in the pH 4 buffer was 64.8±0.9 mg/g support and at the pH 7 - 42.1±1.4 mg/g support. The high loading of the support with biocatalysts causes enzyme crowding on the support surface [38].
According to the obtained results, lipase loading decreased with increasing tested buffers pH.
In the tested model esterification reaction, the highest conversion values were achieved using lipase immobilized at pH 9 - in the system where the lipase loading was the lowest, which probably ensured optimal substrate's availability to the lipase. As lipase loading increased, conversion decreased - possible enzyme aggregation. However, it is assumed that a decrease in conversion values is not only the result of high lipase loading but also the nature (chemical compound) and pH of the buffers used for immobilization.
18) 316 Relation with log P values should be illustrated in Fig. 3
Thank you for your comment. Appropriate changes, as suggested by the reviewer, have been introduced.
Line 374-383
19) 356 Fig. 4, for clarity, explain the meaning of Vis and 75% also in the legend.
Thank you for your comment. Appropriate changes, as suggested by the reviewer, have been introduced.
Line 428-429
The “Vis” means the presence of light in the visible spectral range (400–800 nm), and the 75% means the humidity in the climatic chamber.
20) 395 Is there a way to test this believing?
Thank you for your comment.
After storage (7 days) in the climatic chamber or refrigerator octyl-CALB preserved the catalytic activity (conversion values), almost unaltered. The immobilized lipase demonstrated high insensitivity to storage conditions.
The studies of immobilization of lipases on hydrophobic supports involving the open form of biocatalysts were described, among others, in the paper:
Manoel, E.A.; dos Santos, J.C.S.; Freire, D.M.G.; Rueda, N.; Fernandez-Lafuente, R. Immobilization of lipases on hydrophobic supports involves the open form of the enzyme. Enzyme Microb. Technol. 2015, 71, 53-57, doi:10.1016/j.enzmictec.2015.02.001.
21) 424 Remove first part of sentence.
Thank you for your comment. Appropriate changes, as suggested by the reviewer, have been introduced.
Line 491
The biocatalysts used in the current research, CRL-OF, and CALB, vary in their molecular structure.
22) 429 To compare the catalytic activities of these biocatalysts, CRL-OF was immobilized….
Thank you for your comment. The sentence was modified.
Line 496-497
The lipase from Candida rugosa (CRL-OF) was immobilized onto octyl-agarose support with the use of phosphate buffer (pH 7 and 100 mM).
23) 432 The results with CRL-OF are rather preliminary. Did you also test immobilization of CRL-OF in Tris pH 9? Serendipity cannot be excluded…..
The values of conversion and enantiomeric excess obtained for the CRL-OF were much lower than for the CALB in the tested buffers: pH 4 (citrate), pH 7 (phosphate), and pH 9 (Tris). The use of procedures effective for the CALB does not generate good results for the CRL-OF. We are currently testing the CRL-OF in new systems in search of optimal conditions for this lipase. The results we obtain will be published in another paper.

Round 2
Reviewer 1 Report
Comments and Suggestions for Authors
Authors responded to my comments.
I think that the article can be published.
Author Response
Response to Reviewer 1
Authors responded to my comments. I think that the article can be published.
Thank you for your positive opinion of the manuscript and the suggestion that the article can be published in the journal.

Reviewer 2 Report
Comments and Suggestions for Authors
Fig. 1 is not correct. E1 decreases more rapidly than E2 and both these substrates cannot become bigger in time. The P1 and P2 esters increase in time.
Table 6: lipase loading?
Comments on the Quality of English Language
line 56: One of the most applied group of enzymes are the lipases
line 72: which is converted at a much higher rate....
line 203: was acquired in the pH 9 buffer
line 207: may cause
line 578: using the following equations
Author Response
Response to Reviewer 2
We would like to thank you very much for thoroughly reading the manuscript, issuing an opinion, and presenting valuable insights on quality. The suggested changes were included in the revision.
- Fig 1 is not correct. E1 decreases more rapidly than E2 and both these substrates cannot become bigger in time. The P1 and P2 esters increase in time.
Thank you for your comment. Figure 1 has been corrected as suggested by the reviewer. The text modifications are marked in yellow (line 82-83).
- Table 6: lipase loading?
Thank you for your comment. We are currently working on optimizing the immobilization conditions of CRL-OF on the octyl-agarose support to obtain good values of conversion and enantiomeric excess. Due to the low activity obtained in the organic solvents in the esterification reaction, no further experiments were conducted under these conditions, and lipase loading was not determined. The development of the optimal conditions for the immobilization of this lipase is a challenge for our team. When we reach good results of enantioselectivity in an organic medium, we will present them in the next paper.
Comments on the Quality of English Language
- line 56: One of the most applied group of enzymes are the lipases
Thank you for your comment. The changes suggested by the reviewer have been implemented. The text modifications are marked in yellow.
- line 72: which is converted at a much higher rate....
Thank you for your comment. The changes suggested by the reviewer have been implemented. The text modifications are marked in yellow.
- line 203: was acquired in the pH 9 buffer
Thank you for your comment. The changes suggested by the reviewer have been implemented. The text modifications are marked in yellow (line 205).
- line 207: may cause
Thank you for your comment. The changes suggested by the reviewer have been implemented. The text modifications are marked in yellow (line 209).
- line 578: using the following equations
Thank you for your comment. The changes suggested by the reviewer have been implemented. The text modifications are marked in yellow.
